# Memorial Parking Trees: Resilient Modular Design with Nature-Based Solutions in Vulnerable Urban Areas

**Fortino Acosta *** <sup></sup> **and Stephen Haroon**

Howard R. Hughes College of Engineering, University of Nevada, Las Vegas, NV 89154, USA; haroon.stephen@unlv.edu

*  Correspondence: fortino.acosta@unlv.edu

**Abstract:** Nature-based solutions (NbS) include all the landscape's ecological components that have a function in the natural or urban ecosystem. Memorial Parking Trees (MPTs) are a new variant of a nature-based solution composed of a bioswale and a street tree allocated in the road, occupying a space that is sub-utilised by parked cars. This infill green practice can maximise the use of street trees in secondary streets and have multiple benefits in our communities. Using GIS mapping and methodology can support implementation in vulnerable neighbourhoods. In this research, we based vulnerability assessments for London, Rio de Janeiro, and Los Angeles on the following three indicators: extreme temperature, air quality, and flood-prone areas. Evidence is emerging that disadvantaged populations may live at higher risks of exposure to environmental hazards. The income and healthcare accessibility of neighbourhoods are the two indicators that will help us target these communities for a better and faster decision-making process. The contrast between the results and the 15-min city concept supports our detecting and prioritising neighbourhoods for MPTS implementation, among other NbS solutions integrated into a more inclusive and sustainable urban design.

**Keywords:** nature-based solutions; environmental justice; geographic information systems; urban resilience; green infrastructure; urban design; sustainable urban design; urban vulnerability

## 1. Introduction

In this Anthropocene period, it is necessary to establish a dialogue between urbanism and nature. This conversation could be about Nature-based solutions (NbS). The International Union for Conservation of Nature (IUCN) definition of NbS is a set of actions to protect, sustainably manage, and restore natural or modified ecosystems that address societal challenges effectively and adaptively, simultaneously providing human well-being and biodiversity benefits [1]. This multifunctional attribute is essential to consider when comparing NbS with other traditional solutions designed for a single or narrower function. NbS are a critical component for cities' resilience and regeneration strategies in the face of the climate crisis, and their ability to adapt or restore to an extraordinary extent. Therefore, a resilient city can reduce citizens' vulnerability and the risk of disturbance events, such as hurricanes, floods, droughts, fires, landslides, pollution, and from the social sphere, including health and the economy. In 2020, unequal social-ecological conditions of cities around the world were more evident under the pandemic crisis [2].

Once again, urbanists have to face an environmental challenge and health and social inequalities, exacerbated by city design that allows urban sprawl which prioritises the use of private vehicles, causing more air pollution and deficient use of infrastructure and public transport systems. Since the19th century, we can report large-scale urban interventions to improve the public's health, such as Georges-Eugène Haussmann's renovation of Paris and Frederick Law Olmsted's Emerald Necklace Park in Boston. In both cases, nature has been considered fundamental for the creation of healthier and more livable cities. To date, the incorporation of NbS is widely recognized as sustainable ecosystem management or

the creation of a new ecosystem type that contributes simultaneously to multiple socio-environmental benefits required in every city due to the climate crisis [3]. A reiterative challenge is that NbS are unconsidered in the urban social and economic agenda even though they contribute to most of the Sustainable Development Goals, which in many cases are a recurrent path to address the current climate crisis [4].

In this direction, green streets initiatives have been implemented in underserved neighborhoods to bring physical and social benefits to these communities, as is the case with Portland's Grey to Green infrastructure initiative [5]. Trees, bioswales, and permeable surfaces are among the green infrastructure components that can be applied to streets. Our study focused on trees and bioswale compositions as critical components in extensive built environments that can make a difference from an environmental justice perspective. This unique composition we called parking trees (PT).

This paper aims to analyse PTs' potential on secondary streets in vulnerable areas of three major cities with social inequality concerns and environmental risks, using geographic information systems (GIS). The three selected cities are: London, Rio de Janeiro, and Los Angeles, places in which geographic and social differences are different; however, they share environmental and social stressors associated with their particular spatial patterns [6]. The selection of these cities is based on the following four considerations: known by the authors, defined as a megacity [7], facing sensitive environmental hazards, and with updated open data sites. Along with these criteria, we desired to explore and determine their potential capacity for how a site-specific NbS adapts to complex and contrasting urban realities.

## 2. Literature Review

Cities' cooling measures, such as light color roofs, cool pavements, building insulation, shading structures, and flood control operations optimising pipeline systems, are monofunctional solutions that operate while being isolated from other urban matrix components [8]. Contrarily, vegetation cover on living roofs and streets with trees and bioswales bring solutions for both concerns, plus enhancing air quality, boosting biodiversity, and improving health levels [9]. Health costs are high related to the cost of treating respiratory diseases, cardiovascular diseases, obesity, and mental illnesses, which green space expansions can reduce [10,11]. These NbS in cities are also known as low-impact development, sustainable urban drainage systems, or green or blue infrastructure in different parts of the world. Complementarily, a compact, sustainable growth strategy has to be implemented to limit natural land change, preserve ecosystem services, and increase infrastructure investment efficiency [12]. Therefore, the implementation of new NbS and ecologically-informed planning is necessary to tackle these socio-environmental concerns determinedly.

One of these NbS strategies is using a bioswale variation called Parking Trees, which consists of a bioswale with trees occupying public spaces, predominantly located on roads thought to be used for parking vehicles. These sub-used sites are negative externalities that demean the use of the public space in the city, and adding Parking Trees, in sum, can generate a beneficial impact. As mentioned, a Parking Tree that integrates a bioswale is a component of a sustainable urban drainage system based on ditches covered with a layer of vegetation located between streets and sidewalks or parking lots to capture, temporarily retain, and filter rainwater runoff [13]. The depth of the bioswales ranges from 15 to 30 cm from the vehicular road level. For safety, the bioswale can have a curb with entrances located at specific distances that allow access, along with the distribution of runoff from the vehicular road. Bioswales also make it possible to avoid soil compaction so that trees planted in it will grow faster, and the upheaval of sidewalks by roots will decrease [14]. A perforated drainage pipe connected to the drainage system prevents the water and the substrate from overflowing when storage hits capacity. The maximum absorption time must be 24 h to avoid the presence of mosquitoes, so the type of substrate and the selected plants conditioned to this type of event are essential for its proper functioning [15].

The use of NbS on roads, primary streets, and secondary streets accompanied by permeable pavements, contributing to the removal of polluting particles emitted by motor vehicles, is currently recognised. A plant barrier at vehicles' exhaust height works as the first filter for these heavy particles before spreading to the tree stratum and then to the atmosphere. They also have an essential function for reducing sediments, metals, pesticides, and hydrocarbons that accumulate in runoff caused by rains [16], improving water quality in rivers, wetlands, and other water bodies. On many occasions, space in streets or sidewalks is minimal and limits establishing a continuous ditch; therefore, Parking Trees only replace car parking spaces. Their trapezoidal shape allows adjoining vehicle parking while preserving the width of existing sidewalks. This low-impact development, coupled with a series of feasibility criteria for the election of spaces, could systematically maximize trees on the cities' secondary streets. Parking Trees can be considered part of the IUCN's infrastructure-related approaches category by being a site-specific, low-impact landscape engineering solution that uses nature to tackle socio-environmental challenges and promotes broad participation [2]. For this research, we would like to name them Memorial Parking Trees (MPT), as part of a COVID-19 victims memorial for those citizens who have lost their lives during the current pandemic, emphasizing the link between health and environment and how sustainability measures can restore our relationship with nature.

Three megacities with environmental risks and high social inequality were selected, considering previous knowledge of the city, easy access to previous studies, and contrast in urban morphology. Megacities are dense urban centres with a population of over 10 million people [8]. The affected ecosystem services of these complex urban areas contribute to adverse social conditions. To measure a city's social inequality, we use the Gini Index, with 0 representing perfect equality and 1 representing perfect inequality [17]. London has a Gini coefficient of 0.7 [18], Rio de Janeiro has a Gini coefficient of 0.58 [19], and Los Angeles has a Gini coefficient of 0.5 [20]. The three cities have a population in their city areas of 7.6 million, 6 million, and 4 million, respectively [21]. Finally, the NbS strategy contrasts with a spatial analysis of the 15-min concept city developed by Sorbonne Professor Carlos Moreno, where people are a 15-min walk or cycle ride's reach of schools, supermarkets, parks, work, and other vital areas [22].

## 3. Data and Methods

For this study, the described methods are categorized into five sections using geographic information system tools. The first phase provides neighbourhood mapping based on social inequality. In the spatial type used in London's case, we used the English Indices of Deprivation that quantifies seven distinct aspects of social deprivation [23]. For Rio de Janeiro, we integrate the house income and health index values evenly. For Los Angeles, we used median household income as the only factor in social inequality. The second phase is to create neighbourhood mapping based on the spatial analysis of three environmental risks that are frequent in urban environments: air pollution, extreme heat, and flood-prone areas. The third phase lies in producing a neighborhood vulnerability map by giving an exact weight and summing these together. In this phase, we identify the most vulnerable communities. We select a vulnerable neighbourhood randomly to represent the next stages of the study. The fourth stage identifies how many parking trees we can allocate to make it feasible in the communities. The selection of space follows these field data collection criteria:

1.  Secondary streets selection, no roads or alleys
2.  Do not obstruct driveways
3.  Do not choose space close to existing trees or canopies
4.  Do not obstruct ramps or access to hydrants
5.  Do not select spaces with sewer holes
6.  Selected spaces should have a separation of two vehicles between them
7.  Spaces with some conflicts, such as electric wires and lamp posts, to be indicated

The fifth and last stage describes the areas inscribed within a 15-min city using a one-kilometer buffer spatial analysis from parks, elementary schools, and bus stops. This last analysis shows how this study could complement other urban sustainable initiatives to improve cities' quality of life.

The project considered the International Union for the Conservation of Nature's global standards for NbS, providing information about viability, scalability, and redundancy required for the design of a more resilient city [1]. Results may support the design of new local policies by correlating an environmental justice specific target with an affordable, replicable NbS. This research used the following free spatial data references (Tables 1–3).

**Table 1.** London case study.

| Maps | Source | Spatial Data Type |
|---|---|---|
| London Wards | London Datastore, 2018 [24] | Vector |
| Deprivation Index | Ministry of Housing, Communities & Local Government, 2019 [25] | Statistical table and raster |
| Urban Heat Island Effect Areas [1] | London Unified Model (LondUM), 2011 [26] | Raster |
| Flood Risks Zone | Environment Agency, 2017 [27] | Vector |
| Air Pollution | London Datastore, 2016 [28] | Vector |

[1] Average of the maximum daily temperature across the 2006 heatwave.

**Table 2.** Rio de Janeiro case study.

| Maps | Source | Spatial Data Type |
|---|---|---|
| Rio de Janeiro Barrios (Neighbourhoods) | Data Rio, 2020 [29] | Vector |
| House Income | Data Rio, 2019 [30] | Statistical table |
| Health Index | Índices de Saúde Urbana, 2010 [31] | Raster |
| Urban Heat Island Effect Areas [1] | Data Rio, 2020 [32] | Raster |
| Flood Risks Zones | Prefeitura da Cidade do Rio de Janeiro, 2020 [33] | Data table |
| Air Pollution | Farias, Heitor, 2017 [34] | Raster |
| Hydrology | Prefeitura da Cidade do Rio de Janeiro, 2017 [33] | Vector |

[1] Average of the maximum daily temperature across the 2015.

**Table 3.** Los Angeles case study.

| Maps | Source | Spatial Data Type |
|---|---|---|
| City boundaries for Los Angeles County | Controller Data, 2019 [35] | Vector |
| Neighbourhood Councils | Controller Data, 2019 [36] | Vector |
| Median Household Income per Census Track 2010 in 2018 | Los Angeles Open Data, 2019 [37] | Vector |
| Urban Heat Island Effect Areas | Geotab, 2017 [38] | Raster |
| Flood Risks Zones | Federal Emergency Management Agency (FEMA) Flood Map Service Center, 2017 [39] | Raster |
| Air Pollution | Sierra Andrade, 2019 [40] | Raster |
| Streams and Rivers | Los Angeles Open Data, 2020 [18] | Vector |

## 4. Results

*4.1. Case Study: London, United Kingdom*

The United Kingdom's capital is one of the most important economic and cultural centres in Europe and worldwide. London is one of the most unequal cities in terms of wealth distribution, despite its international prosperity [18]. The English Indices of Deprivation is the Government's primary measure of deprivation for small areas; in this case, we included the 633 wards that divide Greater London's region. The Index of Multiple Deprivation domains are income deprivation, employment and education deprivation, health deprivation and disability, housing and services accessibility, and living environment deprivation and crime [25]. Using 2019 maps and reports, we had to add x, y coordinate data as a layer and assign a homogenous representation color.

With the predicted increase in environmental risks, we selected the most recent maps and information that were available in the Greater London Authority data store, which includes the flood risk zones map produced by the Environment Agency [27], an air pollution map using London Atmospheric Emissions Inventory [24], and Greater London's highest heatwave during summer 2006 [26]. Other studies [41,42] were considered; however, we decided to use the official information provided in these portals. Like the previously mentioned mapping, we homogenised the graphic representation of the existing vector maps, except for the heatwave map, which was only available in a raster form, and a zonal statistics tool was used to define the mean values according to each ward.

Environmental challenges exacerbate the current climate crisis trends. Extreme natural events and other hazards can be accumulated, or present in certain parts of the city. A weight sum tool supports us in generating the next environmental risks map that combines the three environmental problems previously mentioned. A vulnerability map includes social shortfalls and environmental risks together. For this map, we used weight sum, overlaying the Index of Multiple Deprivation with the Environmental risks map created for the city (Figure 1). The areas in brighter red are the most susceptible to manifesting a socio-environmental vexation. The wards in lighter blue are in a safer position than the rest of the city.

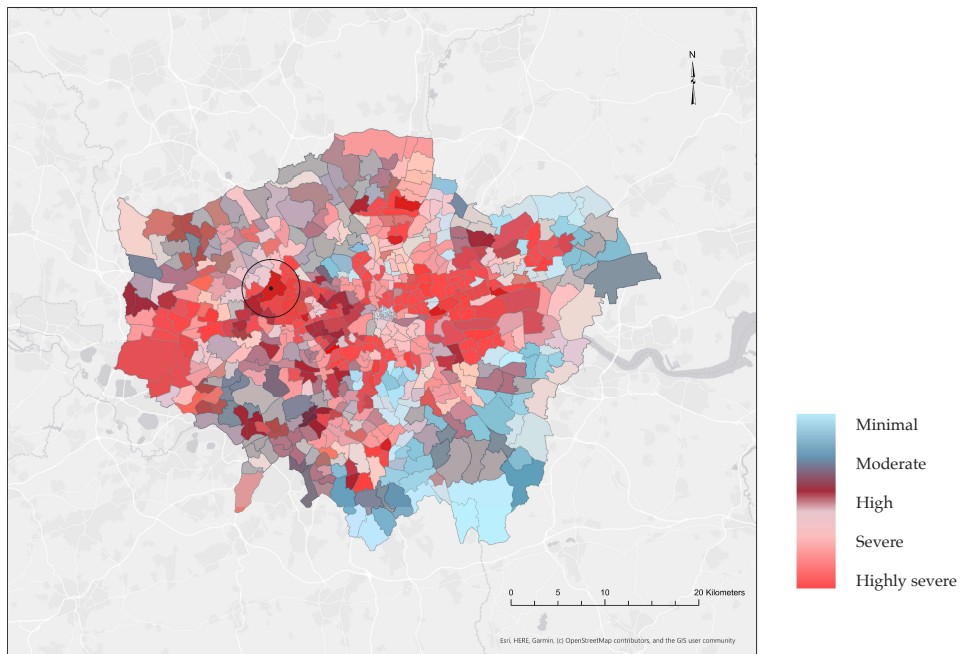

**Figure 1.** Greater London's socio-environmental vulnerability map. Stonebridge is located in the middle of the black circle.

For the MPT spatial analysis, we selected Stonebridge, Northwestern ward, located in the London borough of Brent, three kilometres away from Wembley Stadium. This originally seventeenth-century neighbourhood grew by the river Brent. Even though the environmental risk is considered moderate compared with other areas of the city, the social inequality factor indicates that it is one of the city's most deprived wards. The North Circular Road, rail tracks, and an important industrial area encapsulate this ward. The ward has the condition to allocate 493 MPTs (Figure 2).

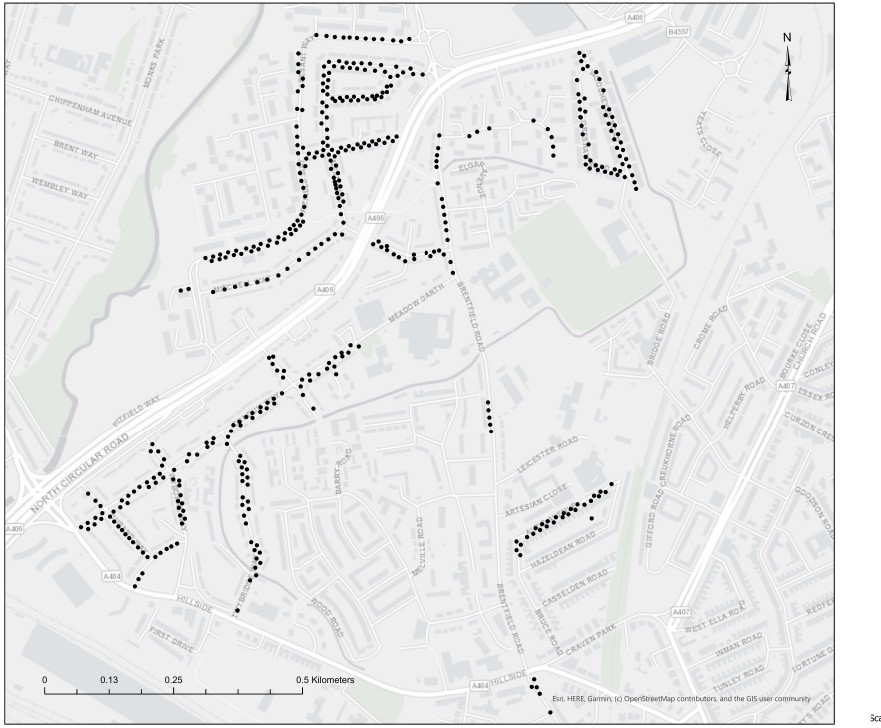

**Figure 2.** A total of 493 MPT spots in secondary streets were identified inside the Ward. Nonetheless, more trees can be located in existing parks, residual spaces, parking lots, and gardens.

The main restrictions to designated vacant spaces in the trees were: positively, an existing tree network is already in place even over the roads. Another limitation was that the industrial park streets are not compatible with large truck maneuvers. Street configuration, especially the use of cul-de-sacs and narrow streets, also prevents more MPT allocations. The MPTs' designs were changed to have a tree occupying one square meter space, protected with a metal tree guard, as consecutive vehicle ramps create an inability to use more road surface space. We suggest including a modular structure for underground infiltration, permeable paving support, and reduction of soil compaction to limit trees' growth.

A 15-min city analysis complements these results (Figure 3). A buffer of one kilometre is calculated for parks, allotments, supermarkets, educational centres, train stations, and bus stops, creating a purple field that shows the area with more walking connectivity. The industrial park is an area that lacks vital destinations. Meanwhile, MPTs add a better experience for pedestrians, and are significantly closer to Brent River Park. Additionally, the visualisation (Figure 4) shows the proposed street intervention.

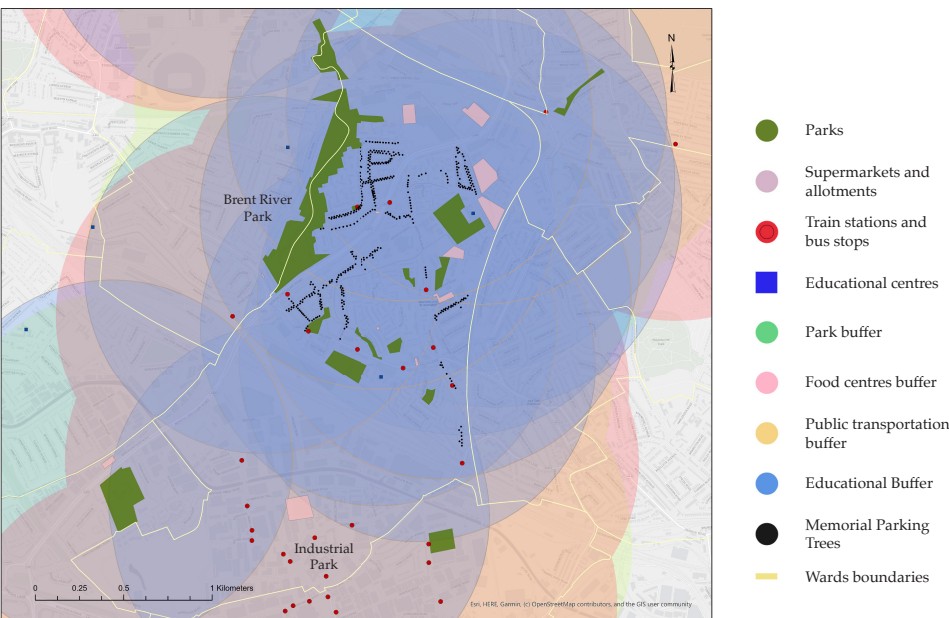

**Figure 3.** Memorial Parking Trees (MPTs) placement and 15-min city analysis inside the boundary of the ward.

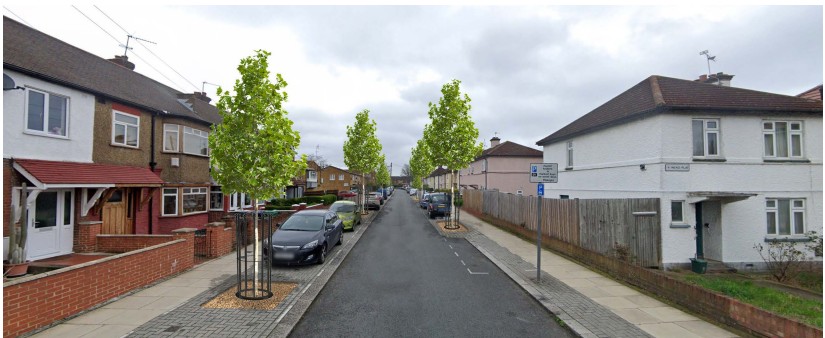

**Figure 4.** Stonebridge MPTs visualization.

### 4.2. Case Study: Rio de Janeiro, Brazil

Once the country's capital until 1960, Rio de Janeiro is one of the world's most vibrant cities. The city, surrounded by Atlantic Rainforest green belts, at the same time faces severe socio-environmental challenges, such as water pollution, segregated urbanisation, landslides, green space accessibility, and air pollution, distributed throughout the 162 neighbourhoods (barrios) that compose the city. The House Income report [30] provided by the local government was the basis for producing the social inequity map, using the add x,y coordinate data tool.

Environmental risks are a constant threat. For the air pollution map, we used the Heitor Farias raster map [34], converting it to neighbourhood polygons by using a zonal statistics tool. A flood risk map was created by joining hydrology maps with flood-prone report tables [33]. Lastly, an average of maximum daily temperatures across the 2015 heatwave raster was used to map neighbourhoods with more extreme temperatures during the warm seasons [32]. Rio de Janeiro's vulnerability map (Figure 5) shows very similar environmental risks, with more mitigated values in the wealthier areas. On the other hand, the Northern area's vulnerability is stressed.

For the MPT spatial analysis, we selected the Barrio de Pavuna, located in the north of the city. Pavuna River is on the northside. On the eastside, an industrial park and an isolated community are contiguous to the Acari River. A rail track divides this in two, and the main motorway crosses through the industrial area. Irregular settlements are scattered

throughout the urban landscape. Tupi tribes and slaves once inhabited the place during the colonial period. Today, the metropolis is embedded in this historic neighbourhood.

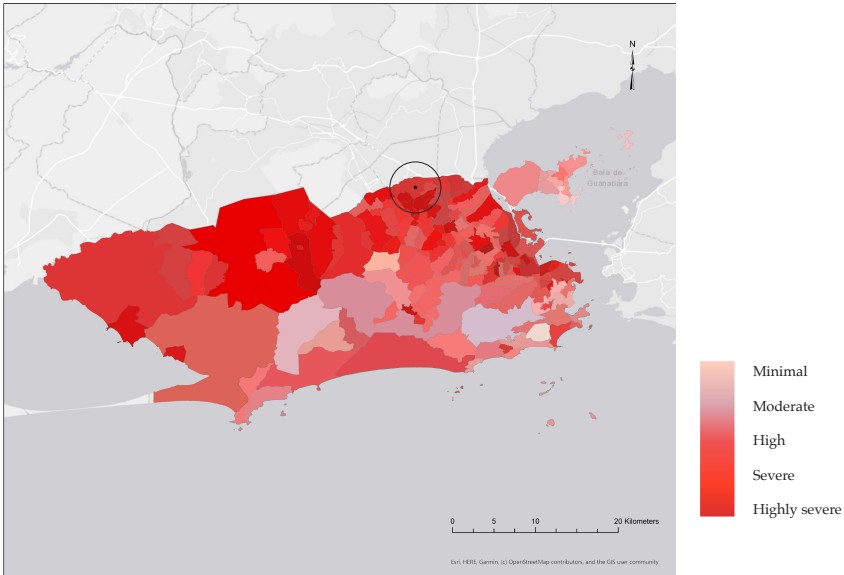

**Figure 5.** Rio de Janeiro's vulnerability map. Pavuna neighbourhood appears in the middle of the black circle.

Extreme temperature and air pollution are part of everyday life. In recent years, several flooding events occurred. The neighbourhood has the condition for the distribution of 1904 MPTs (Figure 6). The main restrictions to the designated vacant spaces for the trees were the industrial park, the irregular settlements, and private communities. Besides, the street configuration, with extraordinarily confined streets, limited the available space; cars have to park over the sidewalks to leave enough space for transit. Most of the streets selected have drainage systems and aerial infrastructure on one side, leaving the other side for the MPTs. The MPTs' design includes a five by 1.5-m bioswale, with a tree and middle stratum, adding more green areas to the city.

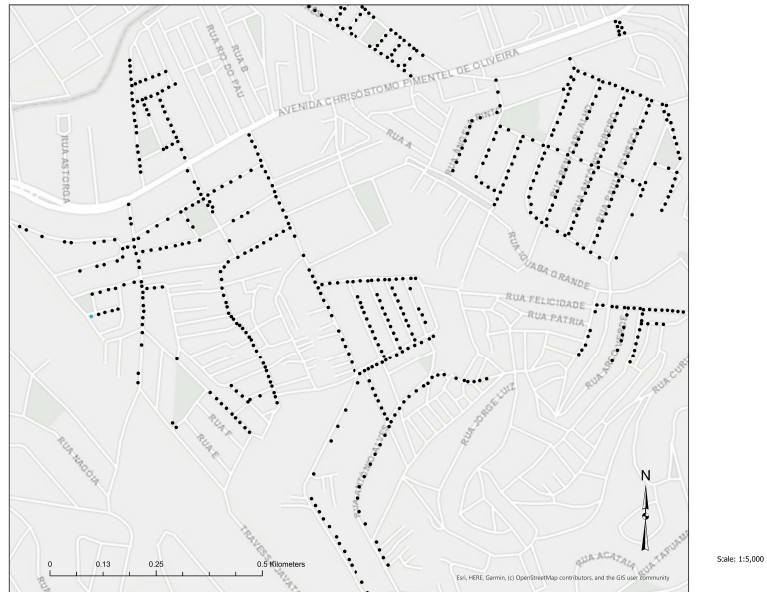

**Figure 6.** Pavuna's secondary streets are very narrow, so MPT's were located only on one side of the streets.

The 15-min city analysis (Figure 7) shows that the industrial park, the community next to it, the insular community located to the north, and the communities settled in the highest elevations decrease the benefits proposed by this urban concept. The visualisation (Figure 8) shows the proposed street intervention on one side of the street.

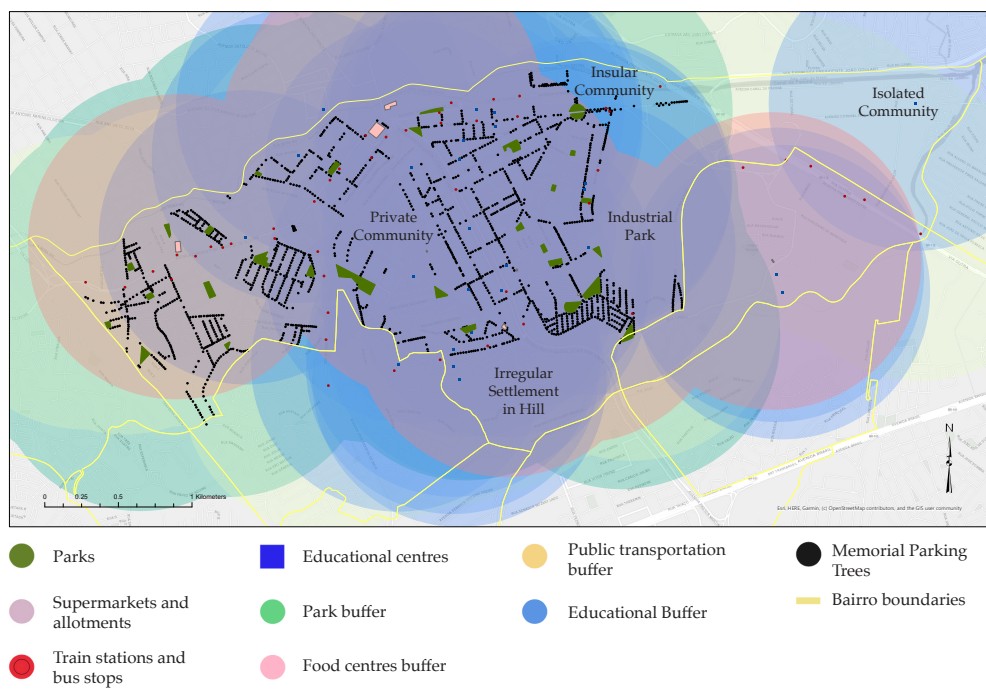

**Figure 7.** A total of 1904 MPT spots were identified within the neighbourhood.

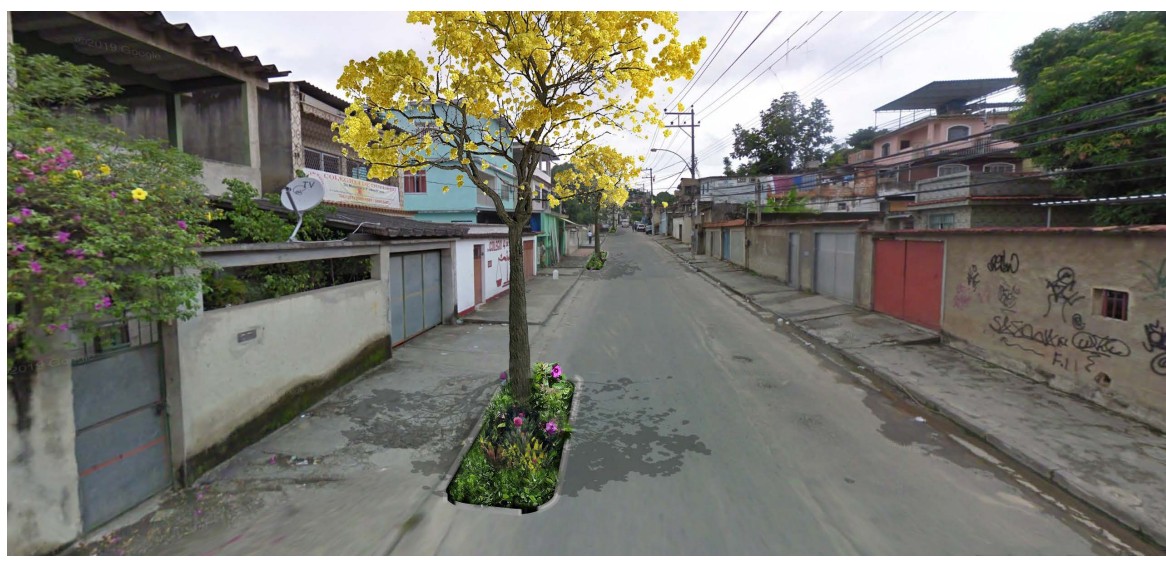

**Figure 8.** Visualisation of Pavuna MPTs.

### 4.3. Case Study: Los Angeles, United States of America

The city of Los Angeles is America's most populous city, situated by the Pacific Ocean. Founded almost 480 years ago (1542) by Spanish conquerors, it has been part of the United States since 1848. Today, the metropolitan area is the third city by gross domestic product worldwide [43]. The Mediterranean weather and topography, with diverse ecosystems such as chaparral, riparian coastal sage scrub, and oak woodlands, are unique characteristics of the region.

For this research, we use Median Household Income per Census Tract 2010 data to represent the social inequality values provided by Los Angeles Open Data platform. For environmental risks, we represent the information provided in the Air Pollution [40], Flood-prone Areas [39], stream and rivers [37], and Urban Heat Island Effect Areas [38] in zip code maps. We created an environmental risk map using a weighted sum tool. Wildfire risk zones are outside of the study area except for the Point Mugu State Park, Malibu Creek State Park, and Topanga State Park corridor on the west side of the city. Due to the urban scope of the research, we do not include them. A vulnerability map (Figure 9) was created by overlaying the social inequality and the environmental risk maps. In this case, we select the southside of the Panorama City neighbourhood for our MPT spatial analysis.

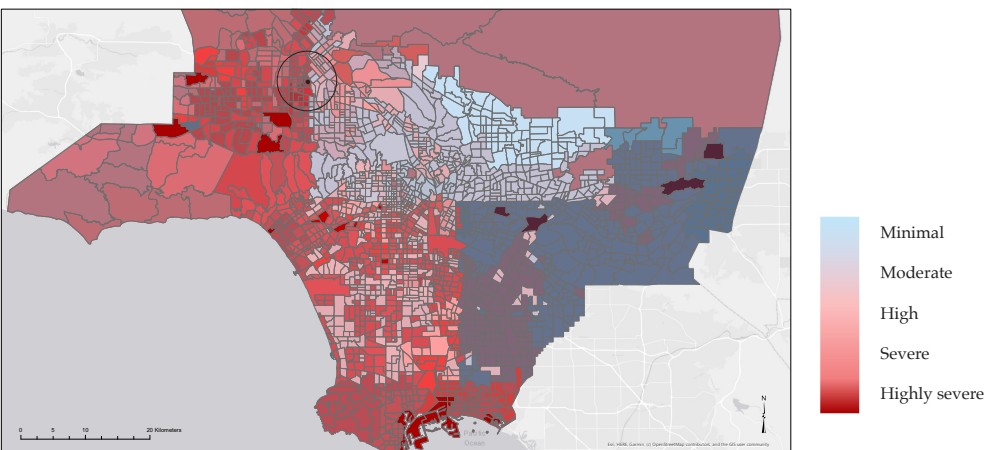

**Figure 9.** Los Angeles' vulnerability map. Panorama City neighbourhood appears in the middle of the black circle.

Panorama City is in the San Fernando Valley, on the north-eastern side of the city, and is the first planned suburb in this area. The neighbourhood features a conspicuous zonification of low-height apartment buildings, single housing, and a large shopping centre. The Pacoima Wash is on the east side of the neighbourhood. For the MPT analysis, we encountered a total of 1501 spaces, an incredible amount, considering that it is known as a suburban community. The spatial analysis noticed that MPT spaces allow the use of both sides of the street. In certain areas, bioswales fuse lawn strips that exist between the road and sidewalk, increasing the benefits of this NbS solution (Figure 10).

By using the 15-min city analysis, we detect that the commercial corridor and an extensive housing continuum carve out the possibility of a walking option within the neighbourhood (Figure 11). The lack of green spaces and schools in this axis is also visible in the study area's north-south poles. Lastly, the visualisation (Figure 12) shows a street using MPTs.

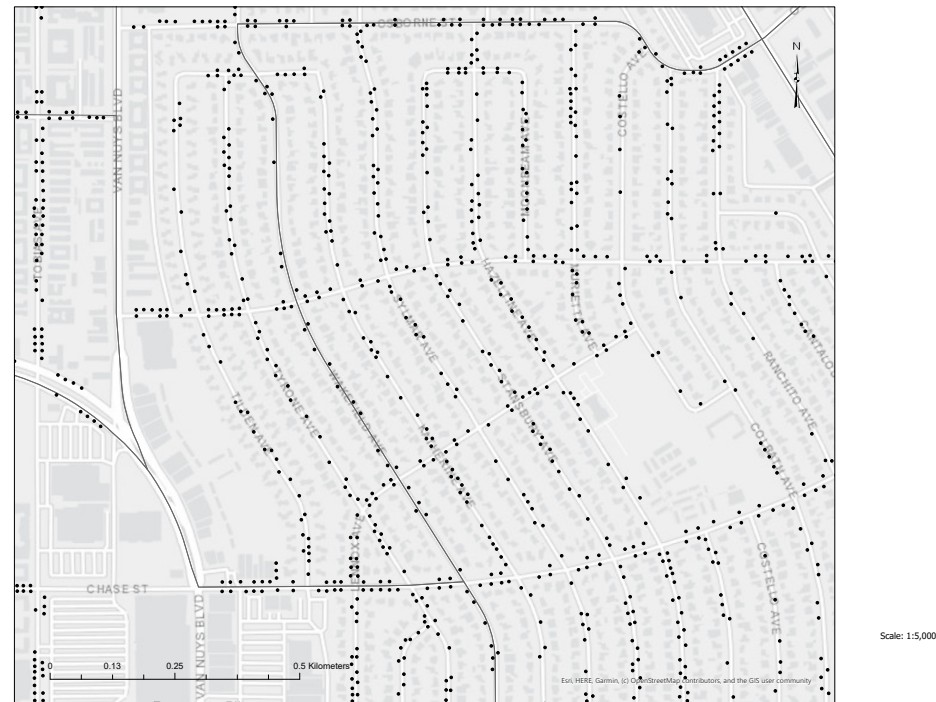

**Figure 10.** Panorama City secondary streets with PMTs, showing the opportunity to increase green cover in the San Fernando Valley.

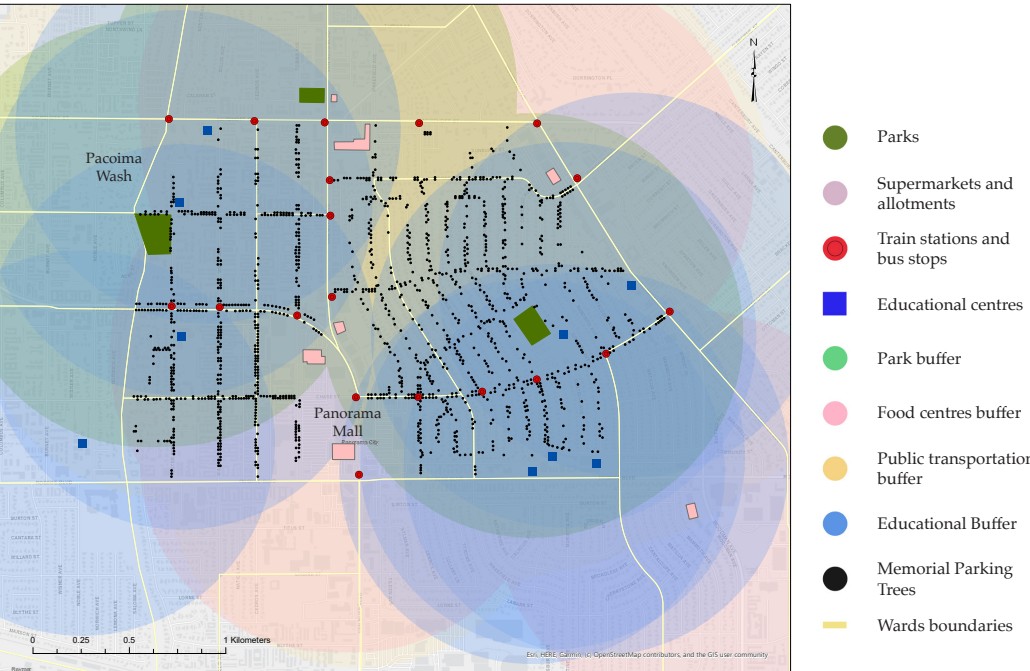

**Figure 11.** A total of 1501 spaces were found in this designed suburban community.

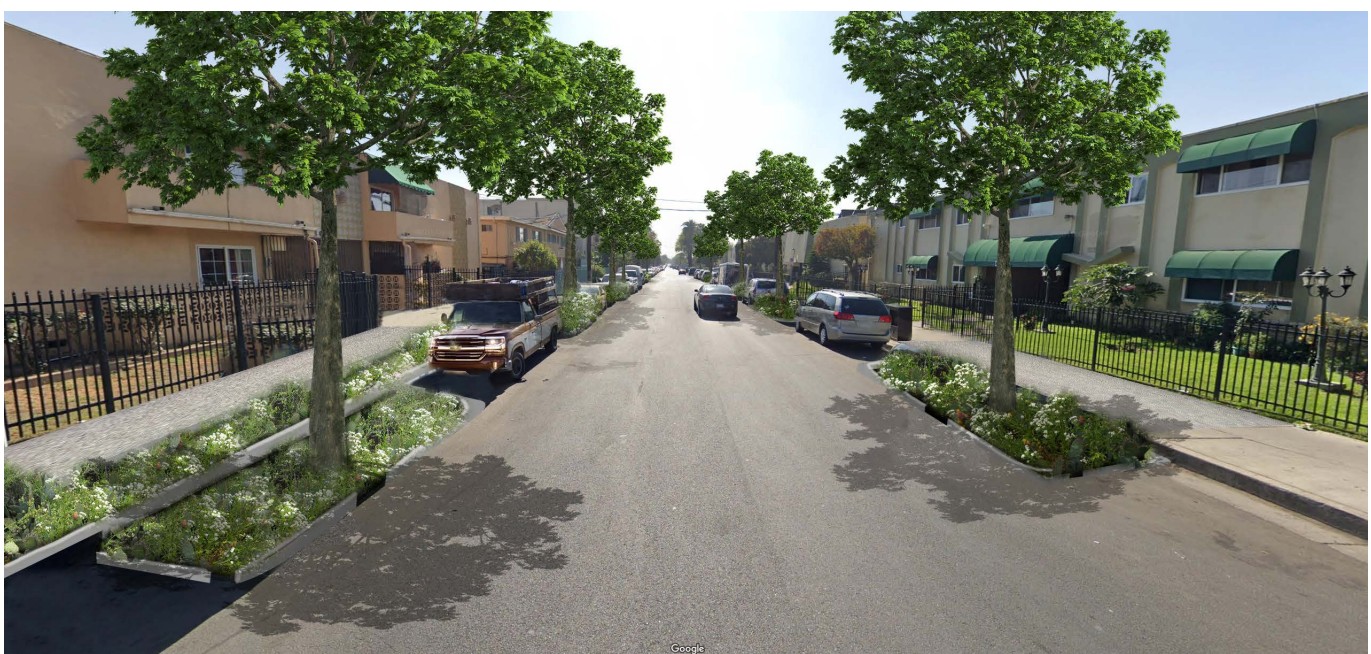

**Figure 12.** Panorama City MPTs' visualization.

## 5. Discussion

NbS cross-sectoral policies required coordinated local environmental, planning, transportation, social, and utility services to improve cities' socio-ecological conditions. Social inequality is expressed differently in the three cities. While in London, it manifests as corridors of deprivation, Rio de Janeiro and Los Angeles are more sectorised, showing government-induced segregation.

The vulnerability analysis shows that most of the environmental risks affect both wealthy and poor neighbourhoods; the difference is that deprived neighbourhoods have fewer tools or opportunities to deal with these environmental hazards. The three case studies show that micro-scale NbS such as MPTs can positively impact vulnerable neighbourhoods independently of the ecosystem. Replicability in MPTs is essential to understand the magnitude of the benefits to be achieved only by replacing fewer than a third of the parking spots available along the roads. This paper shows that thousands of MPTs can mitigate a broad span of the socio-environmental risks mentioned above.

MPTs are one of the easiest NbS that local authorities and citizens can implement. London's case shows that it has an old model for street trees, but this can be improved. MPTs can bring new urban risk considerations that should be assessed, for example, falling under severe weather events, traffic increase, disruption of urban infrastructure by root systems, or maintenance. Thriving secondary street reconfiguration and maintenance requires mutual approval of neighbours and government. Another limitation arises from the limited understanding and acceptance of green infrastructure in existing streets and from specific social behaviors. Perception studies are also required to understand neighbours' receptions of this memorial tribute for lives lost in the Covid-19 pandemic and to understand if people connect the pandemic with environmental justice inequality.

The MPTs' design flexibility allows us to adapt to the condition of each neighbourhood. Other physical restrictions have to be included but were not considered at the beginning of the study, such as street corner visibility, cul-de-sacs, industrial park streets, and very narrow streets. In the end, MPTs have to be part of the solution and not interfere with city mobility. MPTs also have potential in parking lots and industrial parks, but they work better if they are considered for these places at the planning stage. MPTs are not a substitute for preserving natural lands for flood management, but they can mitigate effects of individual events. Additionally, MPTs do not change the broad dynamic of a

city, as noted in the conclusions of the 15-min city, which could helpin optimal land use planning and urban design. However, further research is required to quantify this new MPT network's benefits and compare it with traditional infrastructures.

## 6. Conclusions

From this research, we can conclude that resilient strategies are required to mitigate socio-environmental vulnerabilities expressed throughout our cities. In this direction, MPTs are a minimal expression of an NbS; however, they accomplish the viability, scalability, and redundancy established by the International Union for Conservation of Nature global standards for NbS. A demonstration of a modular systematic MPT approach proved valid and was calibrated for three different urban matrices. The research proves the potential use and replicability of MPTs on secondary streets in other similar or smaller cities beyond the three megacities used for these case studies, but they can also be considered for other urban contexts such as industrial plants, suburbs, semi-suburbs, and parking lots.

MPTs add significantly to the number of street trees that, at the same time, contribute to flood and air pollution reduction in situ. Grey industrial areas, treeless parking lots, cul-de-sacs, one-lane streets, roads, and freeways are built environment components that restrict the use of MPTs and contribute to cities' environmental risks. Meanwhile, industrial areas, open shopping centres, broad roads, private neighbourhoods, and high accidental topographies limit the 15-min city benefits. We recommend that greener industrial areas using NbS, relocation of settlements next to water bodies, redistribution of shops and commerce inside neighbourhoods, and reclamation of residual spaces for green spaces are spatial opportunities for the resilience of cities.

Further MPTs spatial analysis in other neighbourhoods for each city can help us estimate and quantify this proposal's magnitude and potentiality. A pilot project test can result in essential adjustments. Perception analyses are required to consider MPTs as a memorial tribute.

**Author Contributions:** Conceptualization, F.A.; methodology, F.A.; software, F.A.; validation, F.A. and S.H.; formal analysis, F.A; investigation, F.A.; resources, F.A. and S.H.; data curation, F.A. and S.H.; writing—original draft preparation, F.A.; writing—review and editing, F.A.; visualization, F.A.; supervision, S.H.; project administration, F.A.; funding acquisition, F.A. and S.H. Both authors have read and agreed to the published version of the manuscript.

**Funding:** This research received no external funding.

**Data Availability Statement:** Controller Data. City Boundaries for Los Angeles County. Available online: https://controllerdata.lacity.org/dataset/City-Boundaries-for-Los-Angeles-County/sttr-9nxz (accessed on 28 September 2020). Controller Data. Neighborhood Councils (Certified). Available online: https://data.lacity.org/A-Well-Run-City/Neighborhood-Councils-Certified-/fu65-dz2f (accessed on 28 September 2020). Data.Rio. Renda domiciliar. Available online: https://www.data.rio/datasets/indicadores-de-renda-pobreza-pessoas-com-renda-domiciliar-per-capita-abaixo-e-menor-que-r3775-abaixo-e-menor-que-r7550-intensidade-de-linha-de-pobreza-de-r3775-e-de-7550-por-bairros-e-grupo-de-bairros-incluindo-defini%C3%A7%C3%B5es-em-1991-2000 (accessed on 4 October 2020). Data.Rio. Rio de Janeiro Bairros. Available online: https://www.data.rio/datasets/limite-de-bairros?geometry=-45.075%2C-23.138%2C-41.815%2C-22.695 (accessed on 4 October 2020). Data.Rio. Average of the maximum daily temperatures across the 2015 heatwave in Urban Heat Island Effect Areas. Available online: https://www.arcgis.com/apps/MapJournal/index.html?appid=5d9b36b0c4054369b39eb7bf6c2159d7 (accessed on 4 October 2020). Environment Agency. Flood Risks Zones. Available online: https://data.london.gov.uk/dataset/flood-risk-zones (accessed on 21 September 2020). Farias, HS. The spaces of risk to health resulting from atmospheric pollution. Revista de Geografia da UFC. Available online: https://www.researchgate.net/figure/Areas-with-potential-to-accumulate-atmospheric-pollutants-in-the-metropolitan-region-of_fig2_322384260 (accessed on 12 February 2021). FEMA Flood Map Service Center. Flood-prone Areas. Available online: https://static.temblor.net/wp-content/uploads/2017/03/Flood-Map-Los-Angeles.jpg (accessed on 29 September 2020). Geotab. Urban Heat Island Effect Areas. Available online: https://www.geotab.com/heat-in-the-city/#Los%20Angeles (ac-

cessed on 29 September 2020). LA 2050. Income inequality. Available online: https://la2050.org/ (accessed on 18 September 2020). London Datastore. Air pollution. Available online: https://data.london.gov.uk/dataset/london-atmospheric-emissions-inventory--laei--2016-air-quality-focus-areas (accessed on 20 September 2020). London Datastore. London Wards. Available online: https://data.london.gov.uk/dataset/statistical-gis-boundary-files-london (accessed on 20 September 2020). LondUM. Average of the maximum daily temperatures across the 2006 heatwave in Urban Heat Island Effect Areas. Available online: https://data.london.gov.uk/dataset/london-s-urban-heat-island#:~:text=The%20urban%20heat%20island%20(UHI,surfaces%20and%20anthropogenic%20heat%20sources (accessed on 22 September 2020). Los Angeles Open Data. Median Household Income per Census Tract 2010 in 2018. Available online: https://egis-lacounty.hub.arcgis.com/datasets/median-household-income-by-census-tract (accessed on 28 September 2020). Los Angeles Open Data. Streams and rivers. Available online: https://egis-lacounty.hub.arcgis.com/datsetsstreams-and-rivers?geometry=-120.684%2C33.437%2C-115.768%2C34.235 (accessed on 28 September 2020). Ministry of Housing, Communities & Local Government. Indices of multiple Deprivation. Available online: https://data.london.gov.uk/dataset/indices-of-deprivation (accessed on 21 September 2020). Office for National Statistics. Analysing regional economic and well-being trends. Available online: https://www.ons.gov.uk/economy/nationalaccounts/uksectoraccounts/compendium/economicreview/february2020/analysingregionaleconomicandwellbeingtrends (accessed on 21 September 2020). PopulationStat. World Stastical Data. Available online: https://populationstat.com/ (accessed on 18 September 2020). Prefeitura da Cidade do Rio de Janeiro. Flood-prone Areas. Available online: http://g1.globo.com/rio-de-janeiro/noticia/2013/12/prefeitura-mapeia-locais-criticos-de-alagamento-do-rio.html (accessed on 4 October 2020). Sierra Andrade. Air pollution. Available online: https://storymaps.arcgis.com/stories/28e896f8e04a4479a26da75157826569 (accessed on 4 October 2020). Taylor, Jonathon, Wilkinson, Paul, Davies, Mike, Armstrong, Ben, Chalabi, Zaid, Mavrogianni, Anna, Symonds, Phil, Oikonomou, Eleni, & Bohnenstengel, Sylvia I. (2015). Mapping the effects of urban heat island, housing, and age on excess heat-related mortality in London. *Urban Climate*, 14, 517–528. https://doi.org/10.1016/j.uclim.2015.08.001. U.S. Census. *Gini Index*. Available online: https://www.census.gov/topics/income-poverty/income-inequality/about/metrics/gini-index.html (accessed on 18 September 2020).

**Acknowledgments:** We want to thank Marco Martinez for the support to prove that Parking Trees are an NbS feasible solution. Thanks also to Mayra Arzate, Jorge Paiz-Say, and Brian Chernoff for our conversations to link environmental justice, and the commemoration of the people that passed away by the current Sars-Cov-2 pandemic. Finally, we are also grateful to our Technical Writer Meagan Madariaga-Hopkins and to UNLV Libraries, Graduate & Professional Student Association, Howard R. Hughes College of Engineering and the School of Architecture for your funding to publish this research.

**Conflicts of Interest:** The authors declare no conflict of interest.

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
