# Peer review of "Memorial Parking Trees: Resilient Modular Design with Nature-Based Solutions in Vulnerable Urban Areas"

_land, doi:10.3390/land10030298_

Round 1

Reviewer 1 Report

The proposed paper discusses the possibility of the MPT concept implementation in three test sites in London, Rio de Janeiro, and Los Angeles. 
The MPT concept addresses the problem of ensuring sustainable development of urbanised areas by reducing risks through the implementation of a bioswale approach. Bioswale involves the integration of natural and socio-cultural approaches to urban sustainability by incorporating natural elements, such as trees, into the urban infrastructure. The result of the study is to prioritise areas for the implementation of MPT to determine their potential capacity in three testbeds. 
MPT itself is an interesting example of the integration of artificial urban environments, natural objects, and social engineering, which in this case is about increasing public participation in the management of sustainable urban environments through the dedication of individual trees to the victims of the COVID-19 pandemic. As such, the approach laid out in the paper can be considered worthy of discussion. 
The main problem of this study is the inconsistency of the proposed approach. The positive sides of MPT are declared, but their possible negative sides are hardly considered (with the exception of the reduction of parking spaces by a third). At the same time, it is worth considering, for example, the growing threat to traffic connectivity caused by falling trees in hurricanes, for example. Possible numerous other risk factors - such as reduced green electricity production due to solar panels on building roofs being shielded by tree canopies, disruption of urban infrastructure by root systems, the need to raise additional funds for leaf removal in autumn, etc. - should be assessed or at least mentioned. In some cases, the cure can be worse than the disease. 

The work may be published in a journal at least after a more comprehensive consideration of the positives and negatives of the MPT concept.  

Author Response

Dear reviewer,

Thank you for your time in reading the paper and for your useful feedback. 

  • The introduction improved by adding a better description of NbS, adding recent references about their use in underserved neighborhoods, and a better explanation of why we selected those cities (lines 46-55, 63-70).
  • Research Design under interdisciplinary approaches has to open to more experimental methods.  We improved the methodology section by explaining every step and how it can be linked to policies (lines 136-163).
  • We extended the negative impacts and limits of the research (lines 330-336), and it gives a better balance of the positive and negative impacts of this NbS.
  • We agreed that the dication of individual trees to the victims of the COVID-19 pandemic requires research by itself that includes landscape perception research to become valid (lines 337-340).

Reviewer 2 Report

This paper is really inspiring regarding what needs to  be done, now and in the future, to go back to a life closer to nature. The method described shoud be applied and tested in other areas.

Author Response

Dear reviewer,

Thank you for your time in reading the paper and for your beneficial feedback. We improved our English language, extended our introduction, and included several references to improve our literature review. Finally, our discussion has a better balance of positive and negative impacts of using this NbS.     

Reviewer 3 Report

This manuscript addresses topics that are far from my area of expertise in terms of its focus on Nature Based Solutions (NbS) in urban areas to combat increasing environmental hazards, such as air pollution, heat islands, and flooding. Nevertheless, it appears to be a solid first-round effort representing a marginal but useful step forward in understanding the possibilities of Memorial Parking Trees (MPT) as a NbS to urban environmental challenges. The research question is clear, the research approach using GIS data is straightforward and clearly presented, and the visualization scenarios and quantitative findings are intriguing. My main suggestion is to emphasize more clearly the analysis by which this MPT work could be replicated in other urban contexts beyond the three engaging case studies.

Author Response

Dear reviewer,

We appreciate your time in reading the paper and for your useful feedback. 

We improved our English language, extended our method description (lines 155-163), and included the possibilities of being used in other urban contexts (lines 341-344, 358-362). 

Reviewer 4 Report

The paper focuses on a really current and significant urban issue, namely the potential spread of NbS in cities. However, a comprehensive literature review is unquestionably needed since just a few citations are related to scientific works. Moreover, the methodology is not clearly described, and the study area selection is unclear as well. My detailed comments can be seen below:

  • The Introduction in its present form is short; please provide some holistic perspectives regarding urban climate-related challenges and the role of NbS;
  • The first three paragraphs of the Literature review includes only one citation; please increase this number by involving and citing relevant and current studies of the topic;
  • A clear description of the case study selection is completely lacking. London, Rio de Janeiro, and Los Angeles are quite different cities, paying attention to the Gini index and population size. According to these data, the cities cannot be comparable at all regarding their vulnerability. It can be stated that this weakness strongly reduces the scientific quality of the following analyses.
  • The description of the methods applied in the paper is too short and undetailed. How can the authors compare the three cities by using different indicators? How can these indicators be interlinked to provide a general vulnerability overview? What type of vulnerability is projected to be analyzed? The random selection of the analyzed neighborhoods is not a scientific methodology. 
  • The Discussion is nothing to do with the numerous limitations of the study. Furthermore, according to line 310-311, perception studies are projected to scan for collecting information about neighbors’ receptions. How can it be feasible through the randomly selected neighborhoods of totally different cities? 
  • In its present form, the paper seems a thought experiment regarding applying NbS in different urban areas; however, a detailed and precise description of the methodology and policy-oriented aspects is totally missing. Not to mention the assumed impacts of tree planting are not defined.

Author Response

Dear reviewer,

We appreciate your time in reading the paper and for your useful feedback. 

  • The introduction improved by adding a better description of NbS, adding recent references about their use in underserved neighborhoods (lines 46-55, 63-70), and explaining why we selected those cities.  The selection of these cities is based on the following four considerations: Known by the authors, defined as a megacity, facing complex environmental hazards, and, with up-dated open data sites. 
  • Research Design under interdisciplinary approaches requires to open to more experimental methods. Data management is different among cities, not for that, we should refrain from studying them. Our research uses how spatial urban vulnerability looks like on our planet. We only used the Gini Index and population for a brief description of these megacities.
  • The literature review added new citations (72-109). 
  • Unfortunately, numerous vulnerable neighborhoods exist in our cities, there is not "the most vulnerable neighborhood". That was why we continue with the second part of our analysis by using just one of those most vulnerable neighborhoods. Along with these criteria, we desired to explore and determine their potential capacity of how a site-specific NbS adapts to complex and contrasting urban realities.
  • We improved the methodology section by explaining every step and how it can be linked to policies (lines 136-163). 
  • The first paragraph of our Literature Review includes the positive impacts of NbS used. We extended the negative impacts and limits of the research (lines 330-340), and it gives a better balance of the positive and negative impacts of this NbS.

Round 2

Reviewer 1 Report

Manuscript is improved sufficiently. There are some remarks.

132 "Sobornne Professor Carlos Moreno" - maybe "Sorbonne..."?

139 "Indices of Depravation" - maybe "...Deprivation"?

Ready to publish after the correction of observed minor mistyping, if required.

Author Response

Thanks for your review. I corrected the errors mentioned. Also, we reviewed each paragraph with several grammar improvements. Lastly, we corrected the authors' names in the references.

Reviewer 4 Report

The authors made all the corrections I suggested before; therefore, the paper has been largely improved.

Author Response

Thanks for your review. I corrected the errors mentioned. Also, we reviewed each paragraph with several grammar improvements. 
